# COVID-19 Case Management Outcomes Amongst Diabetes and Hypertensive Patients in the United Arab Emirates: A Prospective Study

**DOI:** 10.3390/ijerph192315967

**Published:** 2022-11-30

**Authors:** Aysha Alkhemeiri, Shaikha Al Zaabi, Jeyaseelan Lakshmanan, Ziad El-Khatib, Niyi Awofeso

**Affiliations:** 1Department of Medicine, Tawam Hospital, Abu Dhabi P.O. Box 15258, United Arab Emirates; 2Internal Medicine Department, Mohammed Bin Rashid University of Medicine and Health Sciences, Dubai P.O. Box 505055, United Arab Emirates; 3Biostatistics Department, Mohammed Bin Rashed University of Medicine and Health Sciences, Dubai P.O. Box 505055, United Arab Emirates; 4Department of Global Public Health, Karolinska Institutet, 17176 Stockholm, Sweden; 5School of Health and Environmental Studies, Hamdan Bin Muhammed Smart University, Dubai P.O. Box 71400, United Arab Emirates

**Keywords:** COVID-19, diabetes type 2, diabetes prevalence, outcomes, mortality, United Arab Emirates

## Abstract

The global pandemic of the novel Coronavirus infection 2019 (COVID-19) challenged the care of comorbid patients. The risk imposed by COVID-19 on diabetes patients is multisystemic, exponential, and involves glucose dysregulation. The increased burden for diabetes patients infected with COVID-19 is substantial in countries with a high prevalence of diabetics, such as the United Arab Emirates (UAE). This study aims to explore the prevalence of diabetes, clinical characteristic, and outcomes of patients admitted for COVID-19 treatment with or without a concurrent preadmission diagnosis of diabetes. A prospective study was performed on 1199 adults admitted with confirmed COVID-19 from December 2020 to April 2021 to a single hospital in the UAE. The study compared the demographics, clinical characteristics, and outcomes in COVID-19-infected patients with diabetes to patients without diabetes. The study endpoints include the development of new-onset diabetes, admission to ICU, trends in the blood glucose levels, and death. A total of 1199 patients (390 with diabetes) were included in the study. A diabetes prevalence was detected among 9.8% of the study population. Among the diabetes group, 10.8% were morbidly obese, 65.4% had associated hypertension, and 18.9% had coronary artery disease. Diabetes patients showed higher rates of ICU admission (11.1% vs. 7.1%), NIV requirement (9.6% vs. 6.4%), and intubation (5.45% vs. 2%) compared to the non-diabetes group. Advanced age was a predictor of a worsening COVID-19 course, while diabetes (*p* < 0.050) and hypertension (*p* < 0.025) were significant predictors of death from COVID-19. Nearly three-fourths (284 (73.4%)) of the diabetic patients developed worsened hyperglycemia as compared to one-fifth (171 (20.9%)) of the nondiabetic patients. New-onset diabetes was detected in 9.8% of COVID-19 patients. COVID-19 severity is higher in the presence of diabetes and is associated with worsening hyperglycemia and poor clinical outcomes. Preexisting hypertension is a predictor of COVID-19 severity and death.

## 1. Introduction

The COVID-19 virus emerged as a cause for variable severity pneumonia in Wuhan, Hubei, China [1] at the end of 2019. The causative agent, identified by laboratory research through deep sequencing analysis from the lower respiratory tract samples of infected patients, is a member of the ß-Coronavirus genera [1]. This is the same subgenus as SARS-CoV and MERS-CoV, a class of enveloped positive-sense single-stranded RNA viruses. The virus receptors use angiotensin-converting enzyme 2 (ACE2) to enter the endothelial cells in many organs, including the lungs, heart, kidneys, liver, and nervous system, causing multiple organ damage as a result of critical illness [1,2,3]. This could be explained by the viral infection and inflammatory response produced combined with the host response [4]. The COVID-19 infection induces a proinflammatory response leading to cytokines release and increasing vascular hyperpermeability and multiorgan failure [4,5]. The inflammatory response and associated damage to the lung interstitium induce hypoxia and respiratory failure, leading to potential death [6]. The produced hypoxia due to COVID-19 infection can increase myocardial damage in addition to the direct myocardial injury induced by the virus itself [3,4,6]. Furthermore, hypoxia combined with the immune-mediated response and hypercoagulability leads to neurological complications [4,6], infection-related coagulopathy increasing the risk of thromboembolic events [4,6,7,8] and hepatic injury causing micro-emboli that obstruct the hepatic vasculature [4,6], which is exacerbated with the presence of diabetes. The sepsis-associated inflammatory response leads to kidney injury in COVID-19-infected patients [4,6] and a rapid kidney function deterioration in 30–40% of diabetics compared to nondiabetic patients, as reported by D’Marco et al. [9]. Given the social, healthcare, and economic implications associated with diabetes [10,11,12,13,14], COVID-19 has further increased the burden on the diabetes population, as well as the healthcare services [15].

The meta-analysis revealed a two to three-fold increase in the severity of COVID-19 infection in diabetic patients [16,17,18] and an increase in poor clinical outcomes and mortality [13,19,20,21,22,23,24,25,26,27,28,29]. Shrestha and colleagues [30] reported a mortality rate of 9.3% among non-diabetes patients, 24.96% among patients with new-onset diabetes, and 16.03% among patients who are known to be diabetic. Diabetes is one of the leading causes of morbidity and mortality worldwide [20], where glucose dysregulation and a hyperinflammatory status increase the susceptibility to infection and, in the presence of COVID-19 infection, contribute to severe pneumonia [12], increasing the length of hospital stays and health costs [13,14], increasing the rate of Intensive Care Unit (ICU) admission [12,20,21,29,31,32,33,34,35,36], premature senescence [20,37,38], and the resulting mortality [11,12,20,21,23,32,33,34,38,39,40]. Karagiannidis and colleagues [28] reported that 38.9% of COVID-19-infected diabetics required ventilation in Germany, and the England (UK) survey of 23,804 patients infected with COVID-19 revealed the deaths of 32% of patients with type 2 diabetes and 1.5% with type 1 diabetes [9]. Variable studies have shown the association of diabetes with an increased risk of mortality (22–31%) due to COVID-19 compared to those without diabetes (2–4%) [14,41,42]. New-onset diabetes was detected among COVID-19 patients associated with adverse events in 20.7% of diabetics and 15.3% of nondiabetics [30]. The Chinese CDC [43] shows that the mortality associated with diabetic patients is 7.3% compared to the overall mortality of 2.3%. Barron and colleagues [44] reported 23,804 COVID-19 deaths in which one-third of the patients had diabetes (31.4% with type 2 diabetes and 1.5% with type 1 diabetes). The Odds Ratio of in-hospital deaths with COVID-19 was 3.51 for type 1 diabetes and 2.03 for type 2 diabetes compared to the population without diabetes [44]. Kastora and colleagues [45] reported the mortality rate among type 2 diabetes patients as 20.1% compared to 18.5% in type 1 diabetes and reported the higher mortality among diabetes patients from the Far and Middle East compared to the cases reported in the EU and America. De Almeida-Pititto [46] concluded the association of diabetes with COVID-19 results in a 2.5-fold increase in mortality and a 2.3-fold increase in severity. Another important finding raised by Rathmann and colleagues [47] suggested the increased incidence of type 2 diabetes in patients with COVID-19 after recovery from infection, which could be explained by the association of COVID-19 infection with insulin resistance and new-onset hyperglycemia in patients without previous diabetes [48]. Furthermore, Sharma and colleagues [48] attributed the new-onset diabetes to direct pancreatic cell damage induced by the COVID-19 infection and aggravating the hyperglycemic status, causing new-onset diabetes. Guo et al. [49], in their retrospective case–control study, concluded that, even in the absence of other comorbidities, diabetics with COVID-19 infection are at a higher risk of developing the severe disease and progression towards a worse prognosis.

Some studies have suggested a bidirectional relationship between diabetes and COVID-19, in which the existence of diabetes increases the risk of severe COVID-19, and severe COVID-19 has a diabetogenic effect [30,50,51]. More pertinent is the study of Lima-Martinez and colleagues [52], which proposed an association of some viral infections with the new-onset of diabetes, and a similar theory was applied to COVID-19 infection. Diabetes and hyperglycemia compromise innate and humoral immunity. Diabetes produces an inflammatory status in the body affecting the regulation of glucose and insulin sensitivity that, under COVID-19 infection, might promote the cytokines systemic inflammation response [52]. The expression of ACE2 in the pancreas facilitates the binding of the COVID-19 virus and entry to the beta cells of the pancreas, producing cellular dysfunction and acute hyperglycemia [53].

Even though the prevalence of diabetes in association with COVID-19 infection was similar to the prevalence of diabetes in the general population [24], higher rates of death were seen in the diabetic population during the pandemic. The UAE reported about 40% of deaths linked to diabetes in COVID-19-infected patients [40,54]. This is in line with the previous findings of Yan et al. [55] and Wang et al. [56], whom both reported a susceptibility toward requiring mechanical ventilation and admission to the ICU, as well as higher rates of death from severe COVID-19 in association with older age, the presence of diabetes, and severe inflammatory response compared to those without diabetes. Earlier studies suggested the prevalence of diabetes among mild COVID-19 patients to range from 5.7 to 5.9%, while the prevalence of type 2 diabetes in severe COVID-19 patients was much higher, reaching 22.2–26.9% [55,56]. Furthermore, diabetes is reported as the third-most prevalent underlying comorbidity in COVID-19-infected patients [24,49,57]. Diabetic patients are more likely to develop severe pneumonia [12] due to increased susceptibility to hyperinflammation and cytokine storm development [49], with an underlying impaired immune response secondary to hyperglycemia [22], leading to severe disease. Diabetes mellitus was identified among 14% of COVID-19-infected patients, as reported from the systemic review and meta-analysis by Mahumud and colleagues [22], which indicated that diabetes was overrepresented amongst the COVID-19 patient population but may also indicate low case detection until a COVID-19 diagnosis. The coexistence of diabetes in COVID-19 infection complicates the management of such patients, with additional considerations not only related to disease severity and the escalation of clinical care but also to the medications used to manage diabetes and their interactions with medications used to treat COVID-19 infection.

According to the International Diabetes Federation (IDF) [58], the international prevalence of diabetes is 9.3%, whereas, in the United Arab Emirates (UAE), it reaches around 16.3% [59]. The prevalence of diabetes in the Northern Emirates of the UAE reached 25.1% among the UAE nationals and 19.1% among expatriates [60,61]. Al Awadi et al. [59] reported a higher prevalence of diabetes among Emiratis at 19.3% compared to 12.4% among expats. Recent studies in the UAE showed the prevalence of diabetes among Emirati adult females to be 8.6%, while the prevalence of prediabetes reached 24% [62]. These facts create further challenges, especially for COVID patients, where existing diabetes could impact their mortality and morbidity outcomes in the long term. The COVID-19 viral infection is associated with pancreatic damage, triggering new-onset diabetes [63,64,65] in many individuals, including youths, and 31% infected with COVID-19 were found to be diagnosed with diabetes [66].

The overall reported diabetes prevalence among the COVID-19-infected population ranged from 5.3% to 20% [16,32,43,67,68]. Meta-analysis studies have shown an increase in the prevalence of new-onset diabetes in the presence of COVID-19 infection: 8–19% in China [12,17,57], 11.2% in the US [18], 8.9% reported from a single study conducted in Italy [24], and 27.2% in Spain [69]. While an earlier study by Kumar et al. [18] reported a lower prevalence of 11.2%, the largest database (*n* = 20,982) from the Chinese Centre for Disease Control and Prevention (CCDC) showed a prevalence of new-onset diabetes of 5% in patients with COVID-19 [16,43], while the prevalence of diabetes among COVID-19 (*n* = 122,653) from the CDC USA database was around 10% [16]. A similar prevalence rate of 10% was reported in Spain [70], while a higher prevalence of 17% was reported in Lombardy, Italy [71]. The meta-analysis of Yang and colleagues [72] included seven studies’ analyses and revealed a diabetes prevalence of 9.7% among COVID-19 patients. The France CORONADO study demonstrated that, among COVID-19 patients, 88.5% had type 2 diabetes and 3% had type 1 diabetes, while 3.1% were diagnosed with diabetes at admission [67]. Shrestha et al. [30] showed that new-onset diabetes has a 19.7% prevalence in COVID-19-infected patients. Mahumud et al.’s [22] systemic review and meta-analysis among 202,005 infected patients with COVID-19 reported a diabetes prevalence of 14%, which might indicate an overrepresentation of diabetes among COVID-19 patients or might indicate low-case detection until a COVID-19 diagnosis. Additionally, the increasing prevalence of type 2 diabetes with age may lead to the overrepresentation of diabetes with advanced age and multimorbidity, which, in turn, is likely to overrepresent death with COVID-19 [44]. Another explanation for the high prevalence of diabetes with COVID-19 infection could be attributed to the diabetes population prone to contact with COVID-19 infection that requires hospitalization. Race and ethnicity might play another role in the different ranges of diabetes prevalence, as well as the different hospitalization guidelines and protocols adopted in each country. Shah et al. [73] evaluated the prevalence of diabetes among UAE migrants, including certain ethnicities (Indian, Pakistan, and Bangladesh), and revealed the overall prevalence of diabetes of 10.7% among female migrants compared to 9.3% among male migrants.

At the UAE level, the early cross-sectional study of COVID-19 patients admitted to a single hospital in Dubai reported that 25.1% of admitted COVID-19 patients had diabetes or prediabetes, of which 14.6% were newly diagnosed either with diabetes or prediabetes [25]. Compared to the international prevalence rate, the UAE has shown higher rates of diabetes prevalence among the COVID-19-infected population. In Abu Dhabi Emirate, Abdallah et al^.^ [74] demonstrated that 38% of COVID-19 patients who required ICU admission were diabetics. Hafidh and colleagues’ [23] retrospective study in Dubai confirmed the higher risk of hospitalization amongst diabetic patients infected with COVID-19, where 22% of hospitalized patients required mechanical ventilation. One-third of patients (33.6%) were diagnosed with diabetes on admission, which might reflect the undiagnosed cases of diabetes in the region [23] or might reflect the COVID-19-related injury to the pancreas. Furthermore, Hafidh and colleagues [23] revealed the worse outcomes in the newly diagnosed cases compared to patients with preexisting diabetes and their requirement for mechanical ventilation and a longer hospital stay. As the diabetes prevalence in the study was high, the reported mortality rate was also higher than the reported rate for the UAE [23]. The older age group and male gender in COVID-19 patients are associated with a higher mortality risk in the UAE [40].

It is valuable to consider that the most prevalent comorbidities in COVID-19 patients were found to be hypertension, diabetes followed by cardiovascular disease, and respiratory disease [72]. The majority of patients with COVID-19 have preexisting hypertension [75], which was observed in 15–30% of COVID-19 patients [41,56]. Bhatti et al. [25] revealed hypertension to be present in 64% of the diabetic patients admitted with COVID-19 infection, of whom 10.6% have associated ischemic heart disease. In their retrospective analysis of patients with COVID-19, Shi et al. [35] showed that patients with diabetes had a greater prevalence of hypertension (56.9%), and cardiovascular disease (20.9%) than patients without diabetes (28.8% and 11.1%, respectively). The early observational studies from Wuhan showed severe COVID-19 infection amongst 23.7% of hypertensive patients compared with 13.4% of normotensive patients [76], where 35.8% of hypertensive patients’ conditions are complicated by ICU admission, mechanical ventilation, and death compared to 13.7% of normotensive subjects [42]. Wang and colleagues [56] reported a similarly high prevalence of 31.2% of hypertension among COVID-19 patients, where 58.3% require ICU admission compared to 21.6% of normotensive patients. Guan et al. [42] found that preexisting hypertension was independently associated with severe COVID-19 (hazard ratio 1.575, 95% CI: 1.07–2.32). The previous alarming data clearly demonstrate that hypertension is a risk factor for severe COVID-19 infection and its consequences. Diabetic patients are more prone to have associated hypertension which is a comorbidity considered a high risk for contracting COVID-19 infection [48].

The protective effect of ACE-2 expression is reduced in diabetes, hypertension, and COVID-19 infection [75,77]. SARS-CoV-2 binding to the endothelium of target cells, including blood vessels, is accelerated by the ACE-2 receptors, in which those target cells upregulated by diabetes and hypertension treated with ACE inhibitors or angiotensin II type-I receptor blockers (ARBs), leading to the exacerbation of infection, with COVID-19 increasing the risk of mortality [77]. However, Zhang et al. [78] found that using ACE inhibitors or ARBs in COVID-19 patients was not associated with an increased risk of disease severity or mortality, and the use of ACEI/ARBs might be associated with a lower risk of mortality [78]. This fact supports that hypertensive patients should continue the use of ACEI or ARB medications when infected with COVID-19. Another important fact is that, during COVID-19 infection, viral hijack of the endocrine pathway might alter the blood pressure regulation metabolism and inflammation [75]. Decreasing the ACE2 activity leads to inflammation, vascular permeability, and lung injury, which is associated with a rise in blood pressure [79]. The downregulation of ACE2 inhibitors might sustain arterial hypertension and pulmonary artery hypertension, resulting in systemic hypertension [79] in those with no existing hypertension. This fact highlights the higher rates of hypertension detected in patients with COVID-19 and diabetes.

The characteristics and outcomes of COVID-19 patients in the UAE might not apply to other countries due to differences in demographics and multiethnicities, which challenges the management of diabetic patients with COVID-19. Thus, investigating the interplay between diabetes and COVID-19 and treatment strategies will elucidate further research opportunities in such vulnerable groups. This prospective study aims to determine the prevalence of diabetes in COVID-19 disease and the clinical outcomes amongst hospitalized patients with COVID-19 managed as per the Abu Dhabi Department of Health protocol for patients with COVID-19 in a single hospital in the Abu Dhabi Emirate. 

## 2. Materials and Methods

The study was performed as a prospective single-center cohort study on novel Coronavirus 2019-confirmed infection patients hospitalized in a COVID-19-designated hospital, Al Ain Hospital in the United Arab Emirates. The laboratory confirmed COVID-19 based on the Reverse Transcriptase Polymerase Chain Reaction (RT-PCR) (real-time polymerase chain reaction) assay from a nasopharyngeal swab. Data were collected from patients’ electronic health records for patients admitted to the hospital with COVID-19 infection during the period from November 2020 to the end of April 2021. Patients hospitalized with COVID-19 infection and who had diabetes constituted the case group, while those who were COVID-19-positive without a previous diabetes history constituted the control group. The electronic medical records of hospitalized COVID-19 patients of Al Ain Hospital were used as the main source of data extraction for the study, where the main parameter measured was the patient’s clinical outcomes during the hospital stay and three months following hospital discharge.

Data on patient demographics, comorbidities, illness onset, clinical parameters, and biochemical results were collected. All participants were screened for nCoV-19 infection using the (RT-PCR) assay with a nasopharyngeal swab upon admission. Study participants were selected on admission to the hospital and divided into two groups: those nCoV-19-positive with known diabetes and those who were nCoV-19-positive without a history of diabetes. Participants were assessed on admission for COVID-19 disease severity and progression. The study aimed to evaluate the process and impact of patient management measures implemented to combat the COVID-19 pandemic in the Abu Dhabi Emirate by determining the outcomes related to COVID-19 management among diabetic and nondiabetic patients. The study objectives were: (1) explore the treatment outcomes of a diabetic patient hospitalized with nCoV-19 infection compared with hospitalized COVID-19-infected patients with no previous history of diabetes, (2) ascertain the incidence of the newly diagnosed diabetes patients among the study participants of the control group with no previous history of diabetes, and (3) explore the course of diabetes among diabetics diagnosed with COVID-19 compared with their pre-infection clinical status.

### 2.1. Study Population

The study population groups consisted of adults aged ≥ 18 years admitted to Al Ain Hospital who were diagnosed with laboratory-confirmed COVID-19. The patient age was distributed among five age categories to evaluate the risk of health outcomes based on the study sample ages. The period of the study was from November 2020 to April 2021. The inclusion criteria for the study were adults aged ≥ 18 years, both genders, confirmed to be infected with nCoV-19, and admitted to Al Ain Hospital. Patients could either have a previous diagnosis/history of diabetes Type 1 or Type 2 (for the case group) or have no diabetes history (for the control group). The study excluded patients under the age of 18 years and pregnant women. 

### 2.2. Assessment and Outcomes

The study compared the outcomes within the case and control groups of COVID-19 infection patients. All study populations were assessed upon admission for blood glucose levels and followed up during hospitalization as per the hospital protocol for managing diabetic patients with COVID-19 infection. Their glycemic levels on admission and during the hospitalization period, were followed up, in addition to the interventions and development of disease complications related to the diabetic group and the control group. The nCoV-19 patients on admission were investigated for the inflammatory parameters of the COVID-19 infection and other parameters, including random glucose levels and glycated hemoglobin A1c (HbA1c) levels. Data collection included patient demographics of age and gender, type of diabetes (type I or II), COVID-19 PCR result, length of hospital stays, and intervention used, as well as the change in fasting plasma glucose, postprandial plasma glucose, patient body weight, and the HbA1c level on admission and after 13 weeks post-discharge.

The nCoV-19 patients were treated in Al Ain Hospital based on symptom severity and computed tomography (CT) chest findings with a combination of hydroxychloroquine, Favipiravir, Remdesivir, Camostat, Multivitamins, Tocilizumab, and low molecular weight heparin (LMWH) as required and according to body weight. The study endpoints, including the development of new-onset of diabetes, the need for ICU care, noninvasive ventilation (NIV), and patient death, were compared between the group of nCoV-19-infected patients with known diabetes and the group with no previous history of diabetes. The participants’ consent was waived, as the data collection was based on collecting the study data and information from the electronic medical records of the patients, using numerical identification numbers to retrieve the medical record information and the follow-up data. Communication with study participants was not required, as the data collection was based on retrieving the information from the patient charts.

### 2.3. Statistical Analysis

Data were screened for outliers using descriptive statistics, such as frequency distribution. Continuous variables were presented with the mean ± SD, and categorical variables were presented with the number and percent. Association between the explanatory variables and the outcomes was tested using the chi-square test with Yates correction conservatively. However, if the cell frequencies were less than 5 and the overall table total was about 25 in the subgroup analysis, then we used Fisher’s Exact probability test. The continuous explanatory variables were compared between the presence or absence of outcomes using the Student’s *t*-test or Mann–Whitney *U* test as appropriate. The appropriateness of which test we decided based on the Q–Q plot of the variable and the Kolmogorov–Smirnov test. In order to adjust for the effect of confounders and other known risk variables and to study the effect of diabetes on the outcomes such as NIV need, etc., logistic regression analyses were conducted, and the results presented as Odds Ratios with a 95% CI. In order to study the interaction between diabetes and vaccine, a new variable was created, as each variable—for example, diabetes—is present or not in relation to the vaccine. Logistic regression was used to study the interactive effect, which is the presence of both variables. The goodness of fit of the model was identified using Hosmer and Lemeshow test statistics. The significance of the model was assessed by comparing the likelihood ratios of the models. Data were analyzed using SPSS v24 software.

## 3. Results

The study sample included 1204 patients who were diagnosed with COVID-19 infection and were admitted to a COVID-19-designated hospital in Abu Dhabi-Al Ain City, UAE, from November 2020 to the end of April 2021. The case sample was matching 387 diabetes patients infected with COVID-19 with the control of 817 COVID-19 patients with no history of diabetes. Table 1 presents further demographic characteristics of the study sample. The diabetic patient group included 181 (46.8%) males and 206 (53.2%) females compared to the control group, which included 423 (51.8%) males and 394 (48.2%) females. The ages of patients in this study included adults 18 years and older. The new-onset diabetes was detected among 80 (9.8%) patients, 378 (97.7%) were diagnosed with diabetes mellitus type 2, 5 (1.3%) with diabetes mellitus type 1, and 19 (2.3%) were prediabetic. The commonly listed comorbidities among the diabetic population were hypertension 253 (65.4%), obesity 167 (44.2%), and coronary artery disease 73 (18.9%). Hyperglycemia was the most common observed complication among 284 (73.4%) and 171 (20.9%) of the nondiabetic cohort.

The diabetes group showed 43 (11.1%) admissions to the ICU, 203 (52.2%) requiring oxygen therapy, 37 (9.6%) patients requiring NIV, 21 (5.45%) requiring intubation, and 23 (5.9%) who died. The lengths of hospital stays showed a mean of 11.3 ± 9.3 days among diabetic COVID-19 patients and 7.7 ± 5.7 days among COVID-19 patients without diabetes. The HbA1c level on admission for the diabetes group was 8.19 ± 2.07, the admission random blood glucose (RBG) was 11.23 ± 5.11 mmol/L, the follow-up HbA1c level was 8.61 ± 2.38 mmol/L, and the RBG was 9.60 ± 3.08 mmol/L. The body mass index (BMI) was 30.72 ± 6.90 for diabetes patients upon admission and 32.22 ± 2.37 upon discharge from the hospital, with a three-month follow-up BMI of 33.11 ± 5.48. Table 1 also lists the management upon discharge of the admitted patients and shows that, among diabetic patients, 145 (37.5%) continued their oral hypoglycemic agents, and 20 (5.2%) continued on insulin therapy, while 228 (58.9%) were started on insulin therapy, 66 (17.1%) required an increase in their insulin dose, and 178 (46%) required the replacement of oral hypoglycemic agents by insulin injection therapy. Table 2 shows the management outcomes, where mortality was associated with old age, the existence of diabetes, and hypertension, where older age was associated with an increased need for ICU admission (OR, 1.016; 95% CI, 1.001–1.031), intubation (OR, 1.026; 95% CI, 1.002–1.051), and mortality (OR, 1.048; 95% CI, 1.020–1.076). Mortality was associated with diabetes (OR, 2.223; 95% CI, 0.999–4.946), hypertension (OR, 3.060; 95% CI, 1.149–8.148), and age (OR, 1.048; 95% CI, 1.020–1.076).

Table 3 shows within the diabetic patient group that 51 (13.2%) received the first dose of the vaccination, the second dose was received by 24 (6.2%), and none received the third dose of the vaccination. In patients without diabetes, around 93 (11.4%) received the first dose of vaccination, 63 (7.7%) received the second dose, and 1 (0.1%) received the third dose. Among patients who received a single dose of the COVID-19 vaccine, 9 (6.3%) required admission to the ICU, 70 (48.6%) required oxygenation, 8 (5.6%) required NIV, and 5 (3.5%) required intubation. With the second dose of the COVID-19 vaccine, the patients admitted to ICU were 5 (5.7%), those who required oxygenation were 43 (49.4%), NIV was required for 4 (4.6%), and none of the patients who took the second dose required intubation.

## 4. Discussion

In this study, we compared the demographic and clinical characteristics, as well as outcomes, in COVID-19-infected patients with diabetes compared to patients without diabetes. Our cohort patients showed almost equal gender distribution, but the diabetic group population was older compared to the group with no diabetes. The majority of the diabetic patients in the study sample (185 (47.8%)) were older than 67 years old, whereas the patients without diabetes showed an almost similar distribution amongst different age groups. The primary outcome of our study showed that the prevalence of new-onset diabetes among our population is 9.8%, which is lower than the internationally reported prevalence [18,24,69]. The presence of diabetes and new-onset diabetes are both associated with an increase in the risk of all-cause mortality in hospitalized COVID-19 patients [80], but whether new-onset diabetes increases the future risk of the persistence of diabetes following COVID-19 infection remains a valid research area. The COVID-19 inflammatory effects damaging the pancreatic cells and inducing insulin resistance could underlie the new-onset diabetes rates [48,81,82].

The previous study showed that diabetes and hypertension often coexist [35], and our study demonstrated that the majority of the diabetic cohort (253 (65.4%)) had preexisting hypertension, which is significantly higher as compared to 193 (23.6%) in the nondiabetic group (*p* < 0.001). Yet, preexisting hypertension (76%) was more predictive of diabetes in COVID-19 patients in our study, which was considered as a secondary outcome. Hypertension was presented in 65.4%, and coronary artery disease was presented in 18.8% of patients with diabetes in our study which is consistent with the international studies confirming diabetes and hypertension as common comorbidities associated with COVID-19 [32]. Similar findings were observed by Sourij and colleagues [83], in which hypertension preexisted in 77% of the diabetic group with COVID-19. Similarly, Shi et al. [35] showed that 56.9% of diabetic COVID-19 patients had preexisting hypertension. On the other hand, hypertension was found to be the most frequent comorbidity seen in patients with COVID-19 [67], and Elemam et al. [40] emphasized the association of hypertension with morbidity in COVID-19 patients. Even though the management of hypertension in COVID-19 was not part of our study, it is important to point to the evidence of the benefit of ACE/ARB medication continuation for hypertension during COVID-19 infection in improving clinical outcomes [78,84,85,86]. The percentage of coronary artery disease was 18.9% (73) in the diabetic group, which is significantly (*p* < 0.001) higher than 4.8% (93) in the nondiabetic group. The close link between diabetes and cardiovascular diseases places diabetes patients at an increased risk of cardiac events. Patients who are obese or morbidly obese constitute 179 (48.4%) diabetic patients and 363 (45.1%) COVID-19 patients with no diabetes history. This indicates that nearly half of the cohort sample was obese or morbidly obese. If we further analyze the obesity among the study sample, the diabetic patients with morbid obesity formed 10.8% compared to 7.6% of those with no diabetes. Obesity as a comorbidity was present in both the case and control samples of the study, which reflects the nature of the UAE community, in which obesity is considered ahigh-prevalence noncommunicable disease in the UAE [87]. Previous data were consistent with the high prevalence of diabetes and obesity among the UAE nationals and expats [88]. Diabetes and older age often correlate with comorbidities such as cardiovascular disease, hypertension, and obesity [67]. Furthermore, the prevalence of diabetes increases with age in the general population and in patients with COVID-19 [67], which reflects the older age of the diabetic population in our study.

Patients with COVID-19 and diabetes experience worsening hyperglycemia with COVID infection. Nearly three-fourths (284 (73.4%)) of diabetes patients in our cohort developed worsened hyperglycemia as compared to one-fifth (171 (20.9%)) of non-diabetes patients, who developed new-onset hyperglycemia related to COVID-19 infection. Even though the difference was significant (*p* < 0.001), in this case, it was not diagnostic of new-onset diabetes, as it can be most probably explained by the dysregulation of glucose induced by COVID-19 infection. Hyperglycemia increases the vulnerability of the cells to the inflammatory effect of COVID-19 [25,89] and influences the disease severity in COVID-19 patients [57,90]. Several studies [13,20,21,22,91,92] have proven the presence of comorbidities associated with increased morbidity and mortality among moderate-to-severely infected COVID-19 patients. In our study, disease complications in terms of an increased rate of ICU admission were seen in 43 (11.1%) patients with diabetes compared to 58 (7.1%) patients without diabetes. Similarly, diabetes patients showed a higher need for NIV (37 (9.6%)) compared to (52 (6.4%)) patients without diabetes. Intubation was required in 21 (5.45%) diabetes patients and 16 (2.0%) of those without diabetes. The death rate was five times higher among diabetic patients (23 (5.9%)) compared to (10 (1.2%)) in non-diabetes patients (*p* < 0.001). Our study in general showed that diabetic patients experience a higher severity of COVID-19 disease and progressive worsening outcomes.

When it comes to the management and treatment of patients with COVID-19, it looks from the analysis that, in both groups (diabetic and nondiabetic), patients almost required the same medications apart from Tocilizumab (monoclonal antibody against IL-6), which was required more frequently in the diabetic group and with a significant difference (*p* < 0.001) that could indicate the severity of COVID-19 in diabetic patients. The use of Tocilizumab has positive effects on severe lung disease and the prognosis in COVID-19 patients [32] and is considered in the case of cytokine storms [93]. Having said that, however, Tocilizumab failed to reduce the risk of severe outcomes in the hyperglycemic patients in our study (*p* < 0.009), which was dissimilar to its effects in normoglycemic patients, as mentioned by Marfella et al. [94]. Aziz et al. [95] revealed a lower mortality rate and reduced need for mechanical ventilation in COVID-19 patients where Tocilizumab was used. Stone and colleagues’ [96] double-blinded study failed to detect the benefit or harm of the use of Tocilizumab in COVID-19 patients, which is not part of our study to analyze the impact of Tocilizumab specifically on patient outcomes. On the other hand, Lim et al. [97] showed that a combination of Tocilizumab and Dexamethasone is effective in reducing mortality in severe COVID-19 infections where single therapy fails. However, the study of Lim et al. [97] did not include the effect of diabetes in COVID-19 management with Tocilizumab and Dexamethasone.

Ivermectin, a known antiparasitic drug noted to have anti-inflammatory and antiviral properties against a range of RNA and some DNA viruses, provides significant survival benefits [98]. Ivermectin use was included in the management guidelines of COVID-19 patients in our study, as the risk of death was reduced with its use in the COVID-19 group. In our study, 194 (50.1%) patients in the diabetic group were treated with Ivermectin and almost a similar number 408 (49.9%) in the nondiabetic group. Inconsistency in the evidence of Ivermectin’s benefit in COVID-19 treatment was the result of other studies that showed no evidence of a benefit [31,99,100,101,102,103]. Furthermore, Marcolino et al. [99] supported that Ivermectin has no role in reducing the mortality risk and risk of mechanical ventilation requirement in COVID-19. Our study did not focus on the medications used for COVID-19 treatment, but it is worth mentioning our findings concerning Ivermectin use.

The management outcomes in our study reported higher rates of diabetic patients (43 (11.1%)) who required the ICU and noninvasive ventilation (37 (9.6%)) compared to 58 (7.1%) and 52 (6.4%) in patients without diabetes, respectively. The need for intubation and the death rate were also higher among diabetes patients (21 (5.45%)) compared to the control group (16 (2.0%)), and the significance (*p* < 0.04 and 0.006 consecutively) remained even after age adjustment using logistic regression. In our study, there was further evidence of increased mortality among diabetic patients (23 (5.9%)) with COVID-19 compared to those who were infected with COVID-19 but had no previous diabetes (10 (1.2%)), which supports the impact of comorbidities—specifically diabetes—on worse outcomes in COVID-19 [39] and a prolonged course of hospital stay (11.3 ± 9.3) days when compared to the nondiabetic patient group (7.7 ± 5.7). In general, glucose dysregulation in diabetes and non-diabetes contributes to increasing in morbidity and mortality and the length of hospital stays [14]. In this study, after adjusting for confounders, the data indicated that diabetes is marginally a predictor of death when COVID-19 infection coexists.

The mortality in our cohort study was found to be associated with diabetes (OR, 2.223; 95% CI, 0.999–4.946), hypertension (OR, 3.060; 95% CI, 1.149–8.148), and significantly with age (OR, 1.048; 95% CI, 1020–1.076). Earlier during the pandemic, a death rate of 40% was reported among diabetic patients infected with COVID-19 in the UAE [40,54], which might be attributed to diabetes superimposed by several risk factors, including advanced age and high hypertension prevalence, among the diabetic group that was aggravated by COVID-19 infection [104]. Globally, diabetes, hypertension, obesity, and smoking are found to be the most important public health problems contributing to a higher risk of death, where, during the COVID-19 pandemic, the presence of such comorbidities was found to contribute to 30% of COVID-19 deaths [105]. The COVID-19 severity and death predictors include male gender, age, cardiovascular disease, chronic kidney disease, chronic obstructive pulmonary disease, insulin use, and hyperglycemia [106]. Mahamat-Saleh and colleagues [105] also reported a 54% higher risk of death from COVID-19 in diabetes patients compared to patients without diabetes, a 42% increased risk of death in hypertensive patients compared to the no hypertension group, and a 45% risk in the obese group compared to the nonobese patients. In our cohort, age was a significant predictor of ICU admission (*p* < 0.031) and death (*p* < 0.001), indicating that the disease severity increases with advanced age. Additionally, the existence of diabetes (*p* < 0.05) and hypertension (*p* < 0.025) were also significant predictors of death in COVID-19 patients. Most studies have stated that diabetes doubles the death risk from COVID-19 [14,35,45,49,67] and, similarly, advanced age [23,27,32]. A logistic regression analysis after adjustment for age and comorbidities (Table 2) showed that age significantly predicts admission to the ICU (*p* < 0.031), intubation (*p* < 0.033), and death (*p* < 0.001). Furthermore, comorbidities in terms of diabetes (*p* < 0.050) and hypertension (*p* < 0.025) are significant predictors of death in COVID-19 patients. Our results are consistent with previous research findings where diabetes increased ICU admissions, noninvasive ventilation, and the death rates [12,21,25,31,33,34,36,74].

In our study, we followed the study population after recovery from COVID-19 infection for three months to explore the course of diabetes among COVID-19 patients compared with their pre-infection clinical status. Our study showed that the drop in the HbA1c and RBG levels after three months of COVID-19 infection was not statistically significant in both groups (diabetes and non-diabetes). This result might be attributed to the lack of most follow-up information on patient HbA1c and RBG levels during follow-up visits. Previous studies published by Cromer and colleagues [107] and Laurenzi and colleagues [108] supported the improvement of glycemia and remission of diabetes following the course of COVID-19 infection resolution. Similarly, Rezel-Potts and colleagues [109] confirmed the risk of high new-onset diabetes, which could remain for at least 12 weeks before declining post-COVID-19 infection. 

The worsening of the glycemic status among the diabetes cohort required changes in their diabetes management, where half (58.9%) of diabetics were started on insulin therapy, 66 (17.1%) required an increase in their original dose of insulin therapy, and 178 (46%) converted from oral hypoglycemic agents to insulin therapy to allow for better glucose level control upon discharge. The aforementioned diabetes management changes were significantly higher as compared to the non-diabetes group (*p* < 0.001). On the other hand, it was observed that, among the patients without diabetes, a significant number of patients were started on insulin therapy (199 (24.4%)), while others were started or continued on oral hypoglycemic agents (0.9% and 0.5%). Our data confirmed the worsening of the glycemic status among diabetes and nondiabetic patients with an underlying COVID-19 infection. 

As the vaccination against COVID-19 was initiated during our data collection period, our preliminary observation revealed that no significant association was detected between the vaccine dose (first or second either received or not) and the disease complications in terms of ICU admission, need for oxygen therapy, NIV, and intubation. Our data analysis showed that the subjects who were diabetics and did not have the first dose of the vaccine had about six times more odds to require NIV as compared to subjects who were diabetics but received the first dose of the vaccine after adjusting for their diabetes status (*p* = 0.08). This suggested the fact that the diabetic subjects who did not receive the first dose of the vaccine had a higher risk for NIV need. The subjects who were diabetic and did not have the first dose of the vaccine had about six times more odds of being admitted to the ICU as compared to subjects who were diabetic but had the first dose of the vaccine after adjusting for their diabetes status (*p* = 0.05). The above findings suggest the fact that diabetic subjects who did not receive the first dose of the vaccine had a higher risk of admission to the ICU. The previous data were not to be considered for valid inferences for several reasons, including the small number of vaccinated cases included in the study. The power for this question was not up to 80%. Other reasons are the Sinopharm vaccine was the main vaccine deployed during the study period, where the Sinopharm vaccine showed lower efficacy against severe disease—for example, compared to Moderna [110].

The main detected limitation of our study related to data collection, as many patients lacked follow-up post-discharge information, which limited the follow-up accurate outcome evaluation related to their new-onset diabetes status and HbA1c and blood glucose levels post-COVID-19 infection. The information was lacking as the patients did not show for the outpatient follow-up visit; thus, there were no records detected on the patients’ charts and no follow-up investigations.

## 5. Conclusions

In conclusion, the prevalence of diabetes among the UAE population was lower than the internationally reported prevalence, but it increased the all-cause mortality in hospitalized COVID-19 patients, adding a burden to the future of noncommunicable diseases in the country. The COVID-19 infection worsens the diabetes course during acute infection, leading to poor outcomes and following recovery where hyperglycemic control is required. The optimization of diabetes patient care and management during COVID-19 infection could improve the glycemic status and predicts better clinical outcomes. Our study revealed several implications to policymakers where the consideration of clinical guidelines for the management of COVID-19 patients with diabetes and new-onset diabetes is required. Public education and motivation encourage vaccine uptake to prevent disease severity among the diabetes population. The final implication is for the public where diabetic COVID-19 patients are at a higher risk of disease complications and death; the prompt management and control of glucose dysregulation improves the disease outcomes. Considering the higher risk of cardiovascular disease in association with diabetes, early disease management and lifestyle modification could prevent disease severity. The long-term follow-up of patients with diabetes and new-onset diabetes following COVID-19 is recommended in terms of treatment considerations and health outcomes.

## Figures and Tables

**Table 1 ijerph-19-15967-t001:** Study population demographic data and case management outcomes.

Study Population Sample (N = 1204)	Diabetes (*n* = 387) N (%)	No Diabetes (*n* = 817) N (%)	*p*-Value
Gender			
Male	181 (46.8%)	423 (51.8%)	0.105
Female	206 (53.2%)	394 (48.2%)	
Age in years			
18–34	9 (2.3%)	139 (17.0%)	0.001
35–44	26 (6.7%)	219 (26.8%)	
45–55	77 (19.9%)	187 (22.9%)	
56–66	90 (23.3%)	130 (15.9%)	
≥67	185 (47.8%)	185 (17.4%)	
New-onset diabetes		80 (9.8%)	
Prediabetes		19 (2.3%)	
Diabetes mellitus type 1, *n* (%)	5 (1.3%)		
Diabetes mellitus type 2, *n* (%)	378 (97.7%)		
Body mass index (kg/m^2^) *n* (%) *			
Underweight	1 (0.2%)	20 (2.5%)	0.047
Normal	74 (20.0%)	154 (19.1%)	
Overweight	116 (31.4%)	268 (33.3)	
Obese	139 (37.6%)	302 (37.5)	
Morbid Obesity	40 (10.8%)	61 (7.6%)	
Comorbidities *			
Hypertension	253 (65.4%)	193 (23.6%)	<0.001
Obesity	167 (44.2%)	351 (43.4%)	0.797
Coronary artery disease	73 (18.9%)	39 (4.8%)	<0.001
Interstitial lung disease	4 (1.0%)	7 (0.9%)	0.763
Respiratory disease	29 (7.5%)	58 (7.1%)	0.805
Diabetes complications			
Hyperglycemia	284 (73.4%)	171 (20.9%)	0.001
Diabetes ketoacidosis	2 (0.5%)	NA	0.103
COVID-19 intervention *			
Hydroxychloroquine	1 (0.3%)	NA	0.146
Favipiravir	314 (81.1%)	644 (77.6%)	0.158
Remdesivir	62 (16%)	142 (17.4%)	0.557
Camostat	81 (20.9%)	166 (20.3%)	0.806
Dexamethasone	178 (46%)	409 (50.1%)	0.187
Doxycycline	195 (50.4%)	390 (47.7%)	0.390
Ivermectin	194 (50.1%)	408 (49.9%)	0.951
Tocilizumab	11 (2.8%)	2 (0.2%)	0.001
Bamlanivimab	4 (1.0%)	5 (0.6%)	0.428
Interferon-B	3 (0.8%)	5 (0.6%)	0.745
Thiamine	144 (37.2%)	336 (41.1%)	0.195
Micronutrients (Vitamin C, Zinc, Vitamin D)	378 (97.7%)	790 (96.7%)	0.351
Management outcomes *			
Intensive care Unit (ICU) admission	43 (11.1%)	58 (7.1%)	0.019
Oxygen therapy	203 (52.5%)	419 (51.3%)	0.704
Non-Invasive Ventilation	37 (9.6%)	52 (6.4%)	0.048
Intubation	21 (5.45%)	16 (2.0%)	0.001
Death	23 (5.9%)	10 (1.2%)	0.001
Vaccination against COVID19 *			
First dose	51 (13.2%)	93 (11.4%)	0.370
Second dose	24 (6.2%)	63 (7.7%)	0.502
Third dose	0 (0.0%)	1 (0.1%)	0.491
Length of hospital stay (mean ± SD)	11.3 ± 9.3	7.7 ± 5.7	
HbA1c (mean ± SD) * ^			
Admission	8.19 ± 2.07	6.37 ± 1.46	<0.001
Follow up 3 months	8.61 ± 2.38	5.94 ± 0.66	<0.001
RBG (mean ± SD) * ^			
Admission	11.23 ± 5.11	6.89 ± 2.38	<0.001
Follow up 3 months	9.60 ± 3.08	5.40 ± 0.46	0.071
BMI (mean ± SD) * ^			
Admission	30.72 ± 6.90	30.19 ± 7.02	0.233
Discharge	32.22 ± 2.37	32.09 ± 8.88	0.978
Follow up 3 months	33.11 ± 5.48	30.57 ± 6.73	0.212
Diabetes management on discharge, *n* (%)			
Start Oral Hypoglycemic Agent (OHA)		7 (0.9%)	0.068
Continue on Oral Hypoglycemic Agent	145 (37.5%)	4 (0.5%)	<0.001
Start Insulin therapy	228 (58.9%)	199 (24.4%)	<0.001
Continue Insulin therapy	20 (5.2%)	4 (0.5%)	<0.001
Increase Insulin dose	66 (17.1%)	1 (0.1%)	<0.001
Replace OHA with Insulin injections	178 (46%)	3 (0.4%)	<0.001

BMI: Underweight < 18.5, Normal 18.6–24.9, Overweight 25.0–29.9, Obese 30–39, and Morbid obese ≥ 40. HbA1c: Normal < 5.7%, Prediabetes 5.7–6.4%, and Diabetes ≥ 6.5. A random blood glucose level (RBG) of ≥11.1 mmol/L (200 mg/dL) is classified as diabetes. * *p*-value was calculated per row, because the study participants could select more than one answer per the corresponding question. ^ The SD was less than half of the mean; therefore, the SD was presented instead of the IQR.

**Table 2 ijerph-19-15967-t002:** Adjusted association between age, hypertension, coronary artery disease, and management.

Management Outcomes	Adjusted OR (95% CI) *	*p*-Value
ICU admission		
Diabetes mellitus	1.245 (0.783–1.982)	0.355
Hypertension	1.193 (0.733–1.941)	0.477
Coronary artery disease	1.010 (0.524–1.947)	0.975
Age	1.016 (1.001–1.031)	0.031
Need for oxygen (NC O_2_)		
Diabetes mellitus	1.023 (0.778–1.345)	0.871
Hypertension	0.884 (0.668–1.169)	0.386
Coronary artery disease	0.862 (0.570–1.305)	0.483
Age	1.007 (0.999–1.016)	0.08
Non-Invasive ventilation (NIV)		
Diabetes mellitus	1.266 (0.772–2.077)	0.349
Hypertension	1.116 (0.665–1.872)	0.678
Coronary artery disease	1.027 (0.507–2.077)	0.942
Age	1.013 (0.997–1.028)	0.116
Intubation		
Diabetes mellitus	1.823 (0.878–3.784)	0.107
Hypertension	1.615 (0.735–3.547)	0.233
Coronary artery disease	0.754 (0.276–2.061)	0.583
Age	1.026 (1.002–1.051)	0.033
Death		
Diabetes mellitus	2.223 (0.999–4.946)	0.05
Hypertension	3.060 (1.149–8.148)	0.025
Coronary artery disease	1.148 (0.485–2.720)	0.753
Age	1.048 (1.020–1.076)	0.001

* Adjusted for age, hypertension, and coronary artery disease.

**Table 3 ijerph-19-15967-t003:** Vaccination dose and admission to the ICU, oxygenation requirement, NIV, and intubation.

Complications	Vaccine First Dose	No Vaccine	** p*-Value	Vaccine Second Dose	No Vaccine	** p*-Value
ICU admission	9 (6.3%)	92 (8.7%)	0.324	5 (5.7%)	96 (8.6%)	0.356
Need for Oxygen	70 (48.6%)	552 (52.1%)	0.435	43 (49.4%)	579 (51.8)	0.665
NIV	8 (5.6%)	81 (7.6%)	0.369	4 (4.6%)	85 (7.6%)	0.301
Intubation	5 (3.5%)	32 (3.0%)	0.769	0 (0.0%)	37 (3.3%)	0.085

* *p*-value was calculated per row, because the study participants could select more than one answer per the corresponding question.

## Data Availability

Data are available upon request.

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
