# Peer review of "COVID-19 Case Management Outcomes Amongst Diabetes and Hypertensive Patients in the United Arab Emirates: A Prospective Study"

_ijerph, 2022, doi:10.3390/ijerph192315967_

Round 1
Reviewer 1 Report
The authors describe the bidirectional relationship between covid-19 and diabetes (and hypertension). The paper is well-written and presented. However, there are some minor points to be improved as follows.
1. P.2, Line 94: Grammatical error.
2. P.3, Line 106: Grammatical error.
3. P.5, Line 250: Grammatical error.
4. P.5, Line 255: Grammatical error.
5. P.10, Line 432: Grammatical error.
6. Information in paragraphs 2-8 of “Introduction” repeatedly talks about redundantly epidemiological data. Please make them more concise.
7. In “Materials and Methods”, please give a reason for the given age categories of 18-34, 35-44, 45-55, 56-66, and ≥67 years old in Table 1. A discussion of the results should be also provided.
Details are given in the commented manuscript.

Author Response
- Grammatical errors in P2 line 94 have been changed now and to be found in P3 line 136
- Grammatical errors in P3 line 106 have been changed now and to be found in P3 line 147
- Grammatical errors in P5 line 250 have been changed now and to be found in P5 line 255
- Grammatical errors in P5 line 255 have been changed now and to be found in P5 line 260
- Grammatical errors in P10 line 432 have been changed now to be found in P9 line 401
The word ‘outcome’ was deleted
- Introduction page 2-8 has been reviewed and redundant information has been removed and modified.
- In Materials and Methods: we used the age categories because the patients' population was not young (mean age 49 and median 62).
Reviewer 2 Report
Strengths
· The methodology is sound, and the case-control prospective design sufficiently allows for the observation of study outcomes.
· The study includes sufficient details of the underlying mechanisms and scientific connections between Covid and chronic diseases, including diabetes.
· The rationale for doing the study is well articulated and precise. The authors build a good case for the prevalence of diabetes in the UAE and its implication when combined with COVID
· Matching and adjustment for age and comorbidities appear to be appropriately done.
Weaknesses
· The introduction needs to be more succinct as there is a lot of repetition of ideas and redundancy. The authors have already built their case by the first page, and therefore, much of the information is repeated while adding no new information or evidence. This is the case in the whole paper and should be revised to take care of this.
· Much of the information in the results section is included in the text and the tables. Authors should choose which information to highlight. If information is already on the table, then there is no need to include it in the text. For instance, the Covid-19 interventions administered.
· While it is clear how many patients were included at baseline, it needs to be clarified how many were retained to follow-up and what, if any, imputations were done to correct for missing data.
· Why include the second limitation in relation to hypertension? It is confusing since this Is a secondary outcome and was never the main focus of the study.
· In relation to the above point, there is a need to outrightly state the primary and secondary outcomes to avoid confusion. This study provides relevant and useful information with a wide scope, perhaps the information provided could have best been done in two separate manuscripts.
· Informed consent waiver: It is not clear if the participant's medical information was de-identified before and issued a study ID to allow for follow-up. If this is the case, it should be made clear. Otherwise, accessing patient's identifiable medical data requires consent!
· Conclusion statement: While hypertension emerged as being more prevalent, it now takes precedence in the conclusion statement, and by reading that conclusion (which is what most authors first check), it is not apparent whether the other aims of the study were met or not. Surprisingly the conclusion in the abstract is better articulated in relation to the study's purpose.
Author Response
- The introduction need to be more succinct:
The introduction was reviewed, and redundant information has been removed and modified
- Result section much information in result section is included in the text and tables:
I have made the changes to this section, were the result section includes the important results only
- It needs to be clarified how many patients were retained to follow up and what, if any imputations were done to correct for missing data:
This was one of the study limitations and was mentioned in the study.
- Why include the second limitation in relation to hypertension:
This was removed from limitation section
- There is a need to outrightly state the primary and secondary outcomes:
This has been stated on the modified discussion section
- It is not clear if the participant's medical information was de-identified before and issued a study ID to allow for follow-up:
The participants' consent is waived as the data collection is based on collecting the study data and information from the electronic medical records of the patients using numerical identification numbers to retrieve the medical record information and the follow-up data. Communication with study participants would not be required as the data collection is totally based on retrieving the information from the patient charts.
- Conclusion statement:
The conclusion section was reviewed and modified with appropriate changes as per the reviewer's comment
Reviewer 3 Report
In this paper, the authors compared the demographic and clinical characteristics as well as outcomes in Covid-19 infected patients with diabetes compared to patients without diabetes, and found Covid-19 severity is higher in the presence of diabetes and is associated with worsening hyperglycemia and poor clinical outcomes in UAE. The conclusion is similar to other studies. There are some improvements to this manuscript:
- The words “Abstract”, “Methods”,“Results”,“Conclusion” in abstract section can be removed.
- The word “Hubie” in line 41 should be “Hubei”.
- The introduction part of the paper is too complicated, the author should further refine it and focus on the relevant research which is closely related to the paper.
- The method section is too simple.
- The discussion section of the paper is too long. Some contents may be described in the result section. It is suggested that the author reorganizes the result and discussion sections.
Author Response
- The words ‘Abstract’, ‘Methods’, ‘Results’, and ‘Conclusion’ can be removed:
The words were removed from the abstract section
- The word ‘Hubie” in line 41, spelling error:
Was corrected to Hubei
- The introduction part is too complicated
It was reviewed and modified as per the reviewer's comments
- The method section is too simple:
The method section further analysis information was added, Pleas see the manuscript method section.
- The discussion section is too long and some contents may be described in the result section
The discussion section and result section were both reviewed and modified as per the reviewer's comments
Round 2
Reviewer 3 Report
I think the paper can be accepted in present form.